# Chick Chorioallantoic Membrane (CAM) Assays as a Model of Patient-Derived Xenografts from Circulating Cancer Stem Cells (cCSCs) in Breast Cancer Patients

**DOI:** 10.3390/cancers14061476

**Published:** 2022-03-14

**Authors:** Monika Pizon, Dorothea Schott, Ulrich Pachmann, Rainer Schobert, Marek Pizon, Marta Wozniak, Rafal Bobinski, Katharina Pachmann

**Affiliations:** 1Department of Research and Development, Transfusion Center Bayreuth, 95448 Bayreuth, Germany; dschott@simfo.de (D.S.); upachmann@laborpachmann.de (U.P.); kpachmann@laborpachmann.de (K.P.); 2Department of Organic Chemistry, University of Bayreuth, 95440 Bayreuth, Germany; rainer.schobert@uni-bayreuth.de; 3Department of Cardiac Surgery, Clinic of Bayreuth, 95455 Bayreuth, Germany; pizon.marek@gmail.com; 4Department of Clinical and Experimental Pathology, Wroclaw Medical University, 50-556 Wroclaw, Poland; marta.wozniak@umw.edu.pl; 5Department of Biochemistry and Molecular Biology, University of Bielsko-Biala, 43-309 Bielsko-Biała, Poland; rbobinski@ath.bielsko.pl

**Keywords:** CAM-assay, cCSCs, tumorspheres, PDX, breast cancer

## Abstract

**Simple Summary:**

Circulating cancer cells—and in particular their very rare subpopulation, circulating cancer stem cells (cCSCs)—are responsible for recurrence and metastasis. In this study, we present a novel process in which patient-derived xenograft (PDX) can be harvested on chorioallantoic membrane (CAM) from circulating cancer stem cells. In our opinion, the CAM-based PDX model using circulating cancer stem cells can provide a fast, low-cost, easy-to-use, and efficient preclinical platform for drug screening, therapy optimization, and biomarker discovery.

**Abstract:**

Background: cCSCs are a small subset of circulating tumor cells with cancer stem cell features: resistance to cancer treatments and the capacity for generating metastases. PDX are an appreciated tool in oncology, providing biologically meaningful models of many cancer types, and potential platforms for the development of precision oncology approaches. Commonly, mouse models are used for the in vivo assessment of potential new therapeutic targets in cancers. However, animal models are costly and time consuming. An attractive alternative to such animal experiments is the chicken chorioallantoic membrane assay. Methods: In this study, primary cultures from cCSCs were established using the sphere-forming assay. Subsequently, tumorspheres were transplanted onto the CAM membrane of fertilized chicken eggs to form secondary microtumors. Results: We have developed an innovative in vitro platform for cultivation of cCSCs from peripheral blood of cancer patients. The number of tumorspheres increased significantly with tumor progression and aggressiveness of primary tumor. The number of tumorspheres was positively correlated with Ki-67, Her2 status, and grade score in primary breast tumors. The grafting of tumorspheres onto the CAM was successful and positively correlated with aggressiveness and proliferation capacity of the primary tumor. These tumors pathologically closely resembled the primary tumor. Conclusions: The number of tumorspheres cultured from peripheral blood and the success rate of establishing PDX directly reflect the aggressiveness and proliferation capacity of the primary tumor. A CAM-based PDX model using cCSC provides a fast, low-cost, easy to handle, and powerful preclinical platform for drug screening, therapy optimization, and biomarker discovery.

## 1. Introduction

Breast cancer is the most common neoplasms among women, and it continues to be associated with a high mortality rate despite the increasing number of early diagnoses and improvement of the initial tumor cure rate [1]. Distant metastases remain the leading cause of death in breast cancer patients and they arise from hematogenous spread of circulating tumor cells (CTCs) from solid primary or metastatic tumor lesions [2,3,4]. Recent studies have provided strong support for the cancer stem cell hypothesis, which suggests that many cancers, including breast cancer, are driven by a subpopulation of cells that exhibit stem cell properties. These cells may mediate metastasis and, because of their resistance to radiotherapy and chemotherapy, contribute to disease recurrence and progression in cancer patients [5,6]. As a consequence, eradicating this subpopulation would be critical in order to achieve patient’s cure [7,8].

Interestingly, some experimental findings have highlighted that a small subpopulation of CTCs displays CSCs features (tumor initiating capability), so that they can be considered as circulating cancer stem cells [9,10,11]. A number of non-invasive diagnostic assays are being developed in order to either diagnose patients with metastatic disease and/or to screen patients for relapse after treatment. A new diagnostic tool, developed in recent years by our team, involves measuring extremely rare circulating cancer stem cells in patients’ blood to detect early-stage metastases or cancer recurrence in treated patients. This functional assay is based on the unique property of cancer stem cells to survive and grow in suspension culture [12]. cCSCs were able to form tumorspheres in vitro, confirming that this subpopulation of CTCs represent tumor initiating cells which can potentially grow into metastatic tumors in vivo. To assess tumorigenicity, breast cancer stem cells can be xenotransplanted into the subcutaneous area under the mammary fat pad of immunocompromised female mice [5]. Although they are associated with ethical and financial limitations, this assay is the ‘gold standard’ for assessing CSC tumorigenicity.

In contrast, the chick embryo chorioallantoic membrane (CAM) is a low cost, reproducible, and reliable model used to investigate several functional features of tumor biology—such as the ability of cells to be tumorigenic, to invade, and to metastasize into the embryo [13]. CAM is a highly vascularized extraembryonic membrane, thus providing rich nutrient conditions for the growth of human tumor cells. Cancer cells can be implanted easily, non-invasively, and in an immunodeficient environment because the chick embryo has an immature immune system in the early stages of development when the cells are implanted. Within days, tumor formation occurs and, in the case of aggressive tumors, metastasizing cells can colonize the embryo’s organs via hematogenous metastasis. In human medicine, this model is widely used in preclinical oncology research to study the growth of many types of tumors, including breast cancer [14]. Moreover, it has also been used as a platform to analyze the values of anticancer drugs [15] and radiation therapy [16]. A recent important advance of the CAM assay is that patient tumor samples can be used to transplant on the CAM, enabling the establishment of a patient-derived CAM tumor [17].

In this manuscript, we describe, for the first time, the use of tumorspheres cultured from circulating cancer stem cells of breast cancer patients to establish patient-derived xenografts on the CAM membrane. These results allowed us to propose that the chick CAM model can be a new tool to evaluate breast CSC properties in vivo with the potential to be used in a personalized medicine context.

## 2. Material and Methods

### 2.1. Study Population

The patient population of this study consists of 75 women diagnosed with breast cancer in different stages of disease. Blood samples were drawn into normal blood count tubes with ethylenediaminetetraacetic acid (EDTA) as an anticoagulant and processed within 72 h of collection in accordance with a recently published protocol [3,12,18]. The study was conducted in accordance with the Declaration of Helsinki and was approved by the ethics committee of University Bayreuth (O 1305/1-GB).

### 2.2. Sphere-Forming Assays

Circulating cancer stem cells were cultured together with leukocytes in medium composed of RPMI-1640 supplemented with L-glutamine, HEPES, penicillin/streptomycin, and growth factors such as EGF, insulin, and hydrocortisone at a density of 2 × 10^5^ cells/mL. Cells were incubated at 37 °C in a humidified 5% CO_2_ atmosphere, and the fresh culture medium was added once a week until cells started to form non adherent aggregates. The formation of spheres was confirmed under an inverted microscope (Zeiss, Jena, Germany) and after 14 days collected by gentle centrifugation for further experiments. Tumorspheres cultured from cCSCs were characterized using typical combination of markers for breast cancer stem cells. EpCAM positive tumorspheres were additionally stained for CD24 (clone ML5, mouse anti-human, BD Bioscience, Franklin Lakes, NJ, USA) and CD44 (clone 515, mouse anti-human, BD Bioscience, Franklin Lakes, NJ, USA) PE-conjugated antibody. To confirm tumor stem cell properties of tumorspheres, we used an aldehyde dehydrogenase isoform 1 (ALDH1) enzymatic assay, which quantifies the ALDH1 activity of tumorspheres. We used an ALDEFLUOR assay kit (Stem Cell TechnologiesTM, Vancouver, BC, Canada) according to the manufacturer’s protocol. Tumorspheres expressing high levels of ALDH1 becoming brightly fluorescent could be easily distinguished from single cells because they develop a solid spherical formation with a diameter >50 µm.

### 2.3. CAM Xenografts Model from Circulating Cancer Stem Cells

In 10 cases of breast cancer patients, CAM xenograft experiments were performed. Briefly, fertilized chicken eggs were maintained in a humidified egg incubator (Grumbach, Asslar, Germany) at 37 °C. Protocols were approved by the Ethics Committee of the University Bayreuth. A small window was made in the shell on day 4 of chick embryo development under aseptic conditions. The window was resealed with adhesive tape and eggs were returned to the incubator until day 8 of chick embryo development. At day 8 the tumorspheres cultured from peripheral blood were mixed (1:1) with standard Matrigel (BD Biosciences) basement membrane matrix to prevent cell dispersion, in a total volume of 20 µL and directly seeded on the membrane of CAM through the small window created earlier (Figure 1).

After completion of the study protocol, the embryo was sacrificed by decapitation. The CAM bearing the tumor was then excised with sterilized surgical scissors, fixed with 4% paraformaldehyde for 24 h, and embedded in paraffin.

### 2.4. Histopathology

Paraffin blocks were produced for each CAM tumor. The histological sections were deparaffinized, hydrated in xylene and graded alcohol series and then stained with H&E according to standard protocol. Histological images (40×, 100×, 200×, and 400×) were acquired by light microscope (Olympus BX43, Tokio, Japan) and digitized using a RGB video camera (Olympus DP 20, Tokio, Japan).

### 2.5. Graphs and Statistical Analyses

All statistical analyses were performed using SigmaPlot Version 13.0 (released 3 December 2017, Systat Software Inc., San Jose, CA, USA). A paired or unpaired Student’s *t*-test was used for continuous variables, as appropriate. One-way ANOVA was performed to calculate the differences among multiple groups. A two-sided *p*-value less than 0.05 was considered statistically significant.

## 3. Results

### 3.1. Clinicopathological Characteristics of Breast Cancer Patients

Clinicopathological information is summarized in Table 1. Twenty-six patients (34.7%) had T1, 33 (44%) T2, 9 (12%) T3, and 7 (9.3%) T4. Thirty-five patients (46.7%) had lymph node involvement. Sixteen patients (40%) had ER-positive and Her2-negative breast cancer, 16 (21%) had Her2-positive disease, of which 5 were also ER-positive, and 13 (17%) had triple-negative breast cancer (TNBC). The majority of breast cancer patients had non metastatic disease (52; 69.3%), while 30% of patients had metastatic breast cancer.

### 3.2. Detection of Circulating Cancer Stem Cells

One of our aims was to detect circulating cancer stem cells using tumorsphere-forming assay. To determine the self-renewal and growth potentials of cCSCs, we maintained the suspension cultures. We were able to generate tumorspheres from peripheral blood in 58 (77.35%) cases. Tumorspheres cultured from cCSCs were very compact and the size increased with time, reaching a diameter of 50–100 µm after 21 days (Figure 2a). Tumorspheres were positive for EpCAM and CD 44, and negative or very low positive for CD24 (Figure 2b). The EpCAM and CD44 staining was very heterogeneous in the individual cells within individual tumorspheres. Most of the tumorspheres showed distinct fluorescence for ALDH1 (Figure 2c) typical for breast cancer stem cells.

### 3.3. Tumorspheres Count and Its Association with Clinicopathological Features

We correlated tumorsphere counts with clinicopathological features. The number of tumorspheres was significantly associated with presence and amount of distant metastasis and also aggressiveness of primary tumor. We found that patients with distant metastasis had statistically significantly more tumorspheres compared to patients without spreading to distant organs (median 32.5 vs. 10.0; *p* < 0.01) (Figure 3a). Furthermore, the highest level of tumorspheres was observed in the patient group with multiple metastases (median 61; *p* < 0.001) (Figure 3b). Additionally, our results indicate that the number of tumorspheres correlated significantly with the aggressiveness of the primary tumor. Patients with Her2-neu positive primary tumor had significantly higher levels of tumorspheres compared to patients with negative Her2-neu histology (median 35 vs. 10; *p* < 0.05) (Figure 3c). Additionally, the number of tumorspheres was positively correlated with the proliferation index Ki-67 and histological grade in primary breast tumor. By definition, tumors exhibiting a Ki-67 index of 15% or more were considered to be connected with worse prognosis. Based on this cut-off criterion, 18 out of the 75 patients were shown to have primary tumors displaying highly proliferative potential. Patients with Ki-67 > 15% had significantly higher numbers of tumorspheres compared to those with Ki-67 < 15% (median 34 vs. 8; *p* < 0.01) (Figure 3d). With respect to differentiation status, we observed that patients with G3 primary tumors had significantly more tumorspheres compared to patients with G1 (median 32.5 vs. 0; *p* < 0.01) (Figure 4). In contrast, age, tumor size, lymph node and ER/PR status were not significantly associated with the number of tumorspheres.

### 3.4. Establishing a Breast Cancer Model from Tumorspheres Using the Chicken Egg CAM System

To establish a breast cancer chicken egg model, we used tumorsphere cultured from peripheral blood of breast cancer patients (*n* = 10). We were able to reproducibly generate tumors on CAM from 5 out of 10 breast cancer cases tested. Each patient was characterized separately in Table 2. In the CAM, tumors derived from tumorspheres were able to invade the CAM and to be vascularized (Figure 5a). Histological examination revealed that the CAM tumors contained extensive neoplastic tissue with a well-organized, preserved, and homogeneous structure, clearly distinguishable from the surrounding stroma and membrane border which resembled the morphology of the patient’s primary breast tumor (Figure 5b). The success rate of tumor development on CAM membrane depended on the number of spheres and the aggressiveness of the primary tumor. Patients with a large number of tumorspheres and high Ki-67 index of primary tumor were more likely to develop tumors on the CAM membrane (Figure 6a,b). The estrogen receptor status did not have any influence on tumorsphere formation and CAM-tumor development.

## 4. Discussion

Cancers exist in an extraordinary variety of types and subtypes, making each cancer individually unique. Tumors are heterogeneous and many cancer cell populations with different features are present. Among these tumor populations, cells with properties of stemness, commonly called CSCs, have been described and associated with more aggressive phenotypes [19]. There is strong evidence suggesting that cancer cells with stem properties selectively resist current cancer therapies, indicating the important role that CSCs play in tumor evolution, relapse, and metastasis [20,21]. Interestingly, some experimental findings have highlighted that a small population of CTCs also displays cancer stem cell characteristics (tumor initiating capability) [12,22,23]. Sphere-forming assays are well-described culture methods that have been used for stem cell isolation, identification, and enrichment from different tissues [24,25,26,27]. Starting materials for these cultures have been commercial cell lines, surgical resection specimens, and also peripheral blood with a subset of circulating cancer stem cells [11,12]. The role of cCSCs was investigated in several studies and the results indicated that cCSCs are linked to an unfavorable prognosis in various cancers [28,29,30,31]. In this report, we confirmed our previous findings that circulating cancer stem cells are able to generate tumorspheres and the number of tumorspheres was associated with presence of metastasis [12]. Patients with metastatic disease had more tumorspheres compared to the patients without metastases. Furthermore, patients with multiple metastases had the highest number of tumorspheres. These observations suggest that the number of spheres may reflect the aggressiveness of the tumor and provide additional information about the progression of the disease. We also observed that sphere-formation rates were positively correlated with Her2-neu, Ki-67, and grading of primary tumor. Patients with Her2-neu positive tumor had more tumorspheres in comparison to patients with Her2-neu negative status. Our results are consistent with those of Korkaya et al. [32] who showed that overexpression of Her2-neu increases the population of mammary stem cells as well as their ability to form spheres. Ginestier et al. [33] showed that, in human breast cancers, there was a correlation between Her2-neu amplification and cancer stem cells frequency as assessed by expression of the breast CSC marker ALDH-1.

Ki-67 is universally expressed in proliferating cells, and it is a predictive and prognostic marker for clinical practice in breast cancer patients [34]. The level of Ki-67 determines the categorization of low risk of recurrence and high risk of recurrence. Clinically, the most widely used cut-off for low-risk recurrence is a Ki-67 index below 15% [35]. In our study, the generation of tumorspheres was associated with Ki-67 status in the primary tumor. Patients with Ki-67 >15% had more tumorspheres compared to patients with Ki-67 <15%. Cidado´s et al. [36] demonstrated that Ki-67 was required for maintenance of cancer stem cells but not cell proliferation. However, additional studies are needed to clarify the exact mechanism of how Ki-67 regulates CSC properties.

The histologic grade also is a prognostic factor for breast cancer, regardless of tumor size and the number of involved axillary lymph nodes [37]. In our population, patients with poorly differentiated primary tumor (G3) had significantly more tumorspheres as compared to patients with well differentiated tumor (G1). To our knowledge, there are no data in the literature with respect to the relationship between the number of cancer stem cells and the histologic grade in breast cancer. However, Mohanta et al. [38] showed that patients with poorly differentiated squamous cell carcinoma of the oral cavity had more CSCs, which correlated with poor prognosis in clinical settings.

The CAM assay is a frequently applied model to study cancer cell invasion and metastasis [13]. In our approach we sought to investigate the functional properties of tumorspheres. The CAM model has many advantages. It is cost effective, allows large scale screening and is an easily reproducible in vivo model [39,40]. We show here, for the first time, that tumorspheres cultured from peripheral blood in patients with breast cancer are able to induce tumors on CAM upon application. The efficacy of inoculation using surgical specimens varies widely, ranging from about 45% [41] to much higher rates of 70–80% [42,43,44] depending on the type of tumor. Baccelli et al. [45] identified a population of tumor cells circulating in blood from breast cancer patients that initiates metastasis in a xenograft assay, albeit at a much lower frequency in obtaining xenografts. As shown in the Table 2, five patients (50%) have successfully grown tumors on CAM and the success rate was associated with the number of tumorspheres. Moreover, we suppose that the inoculation rate is also dependent upon Ki-67 status and histologic grade of primary tumor, because only patients with high Ki-67 and low histologic grade have developed tumors on CAM membrane. Histopathological analysis was the key endpoint assay of our experiment, which confirmed that tumors grown on CAM had the original morphological profile of the patient’s tumor. Given the high efficiency of patient derived tumor on CAM in a short 8-day period, it holds great promise as an in vivo platform to pursue pilot drug screening on an individual patient’s tumor. Our model has an advantage over others because the biological material used for CAM-assay is easily and repeatedly available without high risk to the patient. A limitation of our study is the small number of patients who underwent CAM experiment, but further clinical validation for a large-size sample set remains to be done in future work.

Personalized medicine aims to provide cancer patients with tailored treatments, so it is necessary to use patient-derived tumor models to characterize possible treatment options. Currently, tumor organoids [46] or patient-derived xenograft mouse systems [47] are used for this purpose. Based on our success in growing patient derived tumors from tumor cells circulating in peripheral blood in the CAM assay, our model may provide an interesting alternative for investigating the individual tumor biology and a suitable platform for the testing of different drugs in personalized oncology.

## 5. Conclusions

The number of tumorspheres cultured from peripheral blood of cancer patients and the success rate of establishing patient derived xenografts on CAM directly reflect the aggressiveness and proliferation capacity of the primary tumor. The CAM-based PDX model using circulating cancer stem cells could provide an alternative step in translational cancer research to assess experimental and novel strategies for both diagnostic and therapeutic purposes.

## Figures and Tables

**Figure 1 cancers-14-01476-f001:**
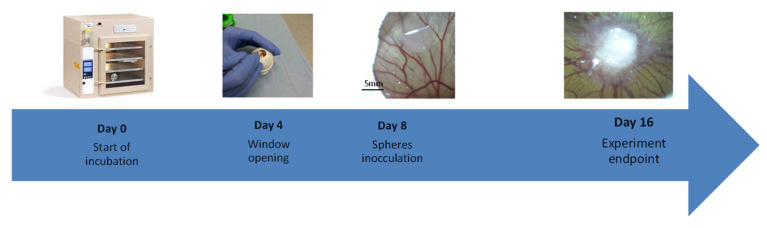
In vivo workflow: A timeline of chorioallantoic member (CAM). Fertilized eggs are incubated for 4 days, at which time a shell window is opened. At embryonic development day 8, tumorspheres are inoculated on top of the CAM. At day 16, the eggs are sacrificed, and the tumor growth is examined.

**Figure 2 cancers-14-01476-f002:**
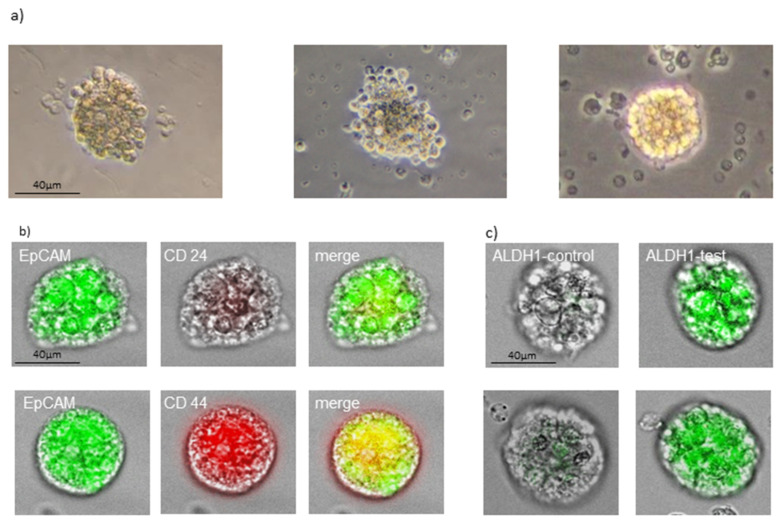
(**a**) Exemplary pictures of tumorspheres in light microscope. (**b**) Fluorescence staining of tumorspheres with cancer stem cell makers typical for breast cancer. (**c**) Expression of aldehyde dehydrogenase 1 (ALDH-1) in tumorspheres. Scale bar = 40 µm; magnification ×200.

**Figure 3 cancers-14-01476-f003:**
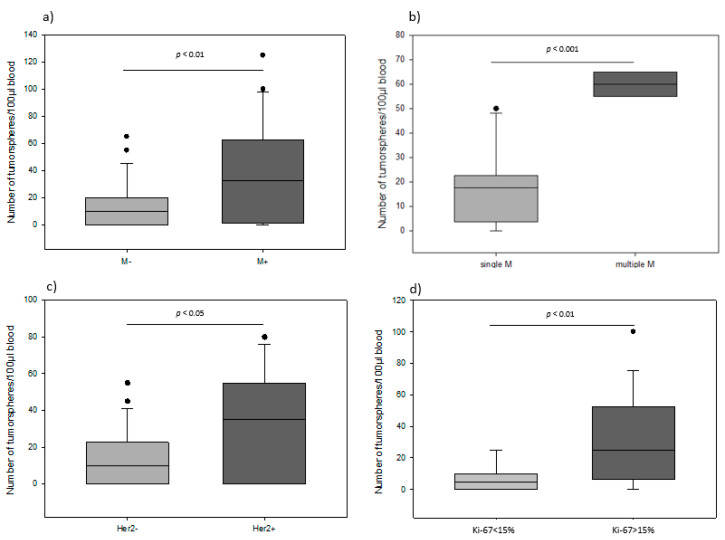
(**a**) Box plot analysis of the number of tumorspheres in patients without and with metastatic disease. (**b**) Box plot analysis showing the number of tumorspheres in patients with single and multiple metastases. (**c**) Box plot analysis of the number of tumorspheres in patients negative and positive for Her2-neu in primary tumor. (**d**) Box plot analysis showing the number of tumorspheres in patients with Ki-67 < 15% and >15% in primary tumor. The horizontal line in box-plot depicts median value. Dots specify outliers.

**Figure 4 cancers-14-01476-f004:**
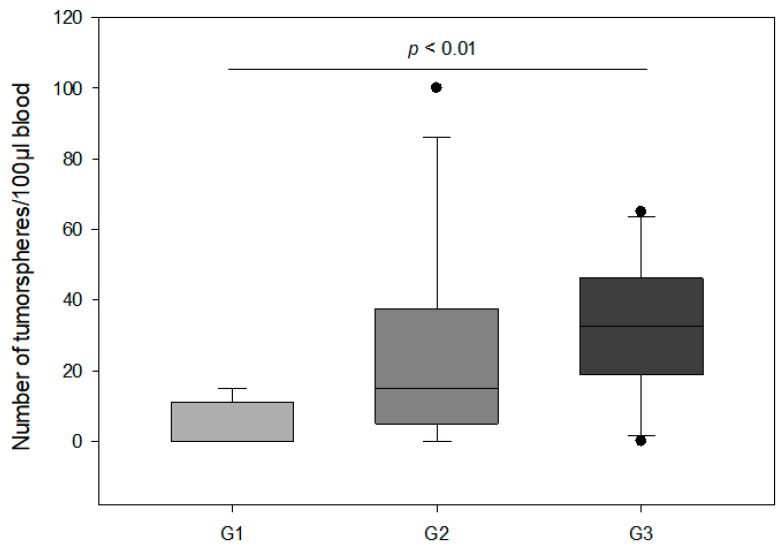
Box plot analysis of the number of tumorspheres according to the histologic tumor grade (G1—well-differentiated tumor; G2—moderately differentiated tumor; G3—poorly differentiated tumor). The horizontal line in box-plot depicts median value. Dots specify outliers.

**Figure 5 cancers-14-01476-f005:**
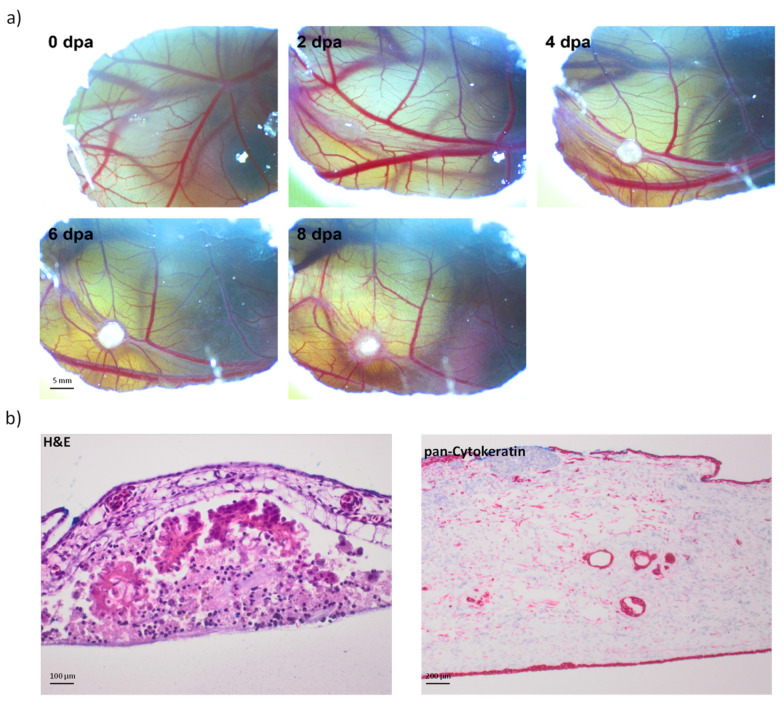
(**a**) Development of a microtumor from tumorspheres of a breast cancer patient on CAM membrane. Scale bar = 5 mm; magnification ×20. (**b**) H&E and pan-cytokeratin staining of a CAM xenograft developed from tumorspheres in light microscopy. Scale bar = 100 µm for H&E, scale bar = 200 µm for pan-cytokeratin; magnification ×200.

**Figure 6 cancers-14-01476-f006:**
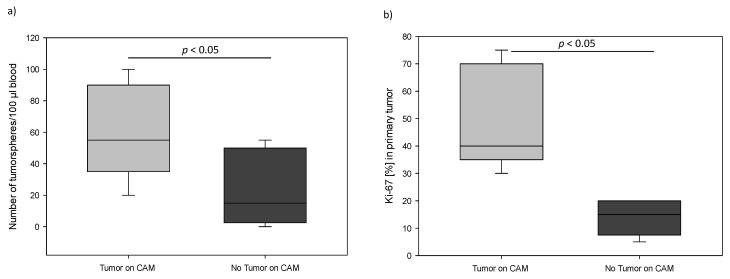
(**a**) Box plot analysis of the number of tumorpheres cultured from peripheral blood in patients who developed a tumor or not on the CAM membrane. (**b**) Box plot showing the Ki-67 proliferation index of primary tumor in patients with breast cancer, with or without tumor on CAM membrane. The horizontal line in box-plot depicts median value.

**Table 1 cancers-14-01476-t001:** Clinicopathological parameters of patients with breast cancer.

Clinicopathological Characteristics	Number of Patients	Median Number of Tumorspheres	*p*-Value
Age			
≤50 years	20	20	*p* = 0.206
>50 years	55	10	
Tumor size			
T1	26	7.5	*p* = 0.318
T2	33	12.5	
T3	9	25	
T4	7	30	
Lymph node			
positive	35	10	*p* = 0.967
negative	40	10	
Clinical stage			
I	13	5	*p* < 0.01
II	23	10	
III	16	10	
IV	23	32.5	
ER/PR status			
positive	62	15	*p =* 0.955
negative	13	20	
Her2-neu			
positive	16	35	*p <* 0.05
negative	59	10	
Grading			
1	14	0	*p* < 0.01
2	33	5	
3	28	32.5	
Ki-67 Index			
>15%	29	34	*p <* 0.01
<15%	46	8	
Metastasis			
positive	23	32.5	*p <* 0.01
negative	52	10	

**Table 2 cancers-14-01476-t002:** Characteristics of patients who underwent CAM experiment.

Patients	Grading	Ki-67	ER/PR	Her2-neu	Metastasis	Tumorspheres	Tumor on CAM
1	G3	40%	positive	positive	positive	80	present
2	G3	75%	positive	negative	negative	100	present
3	G3	65%	negative	negative	negative	50	present
4	G2	40%	positive	positive	negative	55	present
5	G3	30%	positive	positive	negative	20	present
6	G2	15%	positive	positive	negative	55	absent
7	G3	20%	positive	negative	negative	45	absent
8	G2	5%	positive	negative	negative	5	absent
9	G3	20%	positive	negative	negative	15	absent
10	G2	10%	positive	negative	negative	0	absent

## Data Availability

The data presented in this study are available on request from the corresponding author.

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
