# Peer review of "Chick Chorioallantoic Membrane (CAM) Assays as a Model of Patient-Derived Xenografts from Circulating Cancer Stem Cells (cCSCs) in Breast Cancer Patients"

_cancers, 2022, doi:10.3390/cancers14061476_

Round 1

Reviewer 1 Report

It is a very interesting work with possible clinical applications that perhaps would be more detailed. Some minor considerations like; -line 16, an acronym is missing ( circulating cancer stem cells) -line 48-54- 2008 and 2010 is not recent, there is more current bibliography. - Material and methods. A better description of the population, stage, age, hormonal status that I understand may affect the study would be necessary, although the types of tumors are detailed later. Women with hormone-dependent tumors, I understand the aggressiveness of the tumor depends largely on the presence of estrogen and in this model, this factor is not considered, how would this bias be solved? - Line 96, Data of the local ethics committee to which it refers (document number) Finally, the experiment is carried out on 10 patients, their sample being much higher. How have the patients been selected or based on what parameters? It's not quite clear. Perhaps a scheme that determines the sequence of action and development of the work would help. It's a good work.

Author Response

Dear Reviewer,

Thank you very much for your valuable comments, which contributes to the quality of our work.

1) an acronym for circulating cancer stem cells is included.

2) current bibliography was added to the manuscript

3) age and clinical stage have been included into the table of the clinicopathological characteristics. 

4) the name of the ethics committee and document number were added to the manuscript.

5) the estrogen receptor status did not have  any influence on tumorsphere formation and CAM-tumor development( result added in the manuscript). 9 of 10 patients were estrogen receptor positive an influence od this property could not be observed in contrast Her2 positivity seemed to have tendency to have an impact. 

6) there was no selection of patients for CAM-assay experiment. First of all, tumorsphere formation could be observed in 58 patients. Furthermore,  during validation of the assay, there were problems with the viability of the embryo and reproducibility of the results. 

Best regards

Monika Pizon

Reviewer 2 Report

Interesting topic.

Please revise the tables and figures. The are some misspelled words: eg. "positiv", "negativ" in Table 2. Also, some figures are difficult to read because the quality of the photos is poor. 

Author Response

Dear Reviewer,

thank you for your comments. The misspelled words in Table 2. were corrected. 

The quality of all figures were optimized.

Best regards

Monika Pizon

Reviewer 3 Report

The study is well designed and well-founded. Reading is clear and objective. the conclusions meet the proposed objectives.
However, they should improve the quality of images, graphics and their captions.
The writing quality is far superior to the figures.

Author Response

Dear Reviewer, 

Thank you for your comments.

the quality of images and graphics was optimazed.

Best regards

Monika Pizon